# Diagnostic Modality Influences Tuberculosis Detection in People Living with HIV: Eight Years of Data from a Thai Referral Center

**DOI:** 10.3390/diagnostics15182327

**Published:** 2025-09-14

**Authors:** Wannarat Pongpirul, Phanupong Phutrakool, Krit Pongpirul

**Affiliations:** 1Bamrasnaradura Infectious Diseases Institute (BIDI), Department of Disease Control, Ministry of Public Health, Nonthaburi 11000, Thailand; wannaratpa@gmail.com; 2Chula Data Management Center, Faculty of Medicine, Chulalongkorn University, Bangkok 10330, Thailand; phanupong.dell@gmail.com; 3Center of Excellence in Preventive and Integrative Medicine (CE-PIM), Faculty of Medicine, Chulalongkorn University, Bangkok 10330, Thailand; 4Clinical Research Center, Bumrungrad International Hospital, Bangkok 10110, Thailand; 5Department of Infection Biology & Microbiomes, Faculty of Health and Life Sciences, University of Liverpool, Liverpool L69 7ZX, UK

**Keywords:** HIV, tuberculosis, TB/HIV co-infection, diagnostics, GeneXpert, chest radiography, screening algorithm, Thailand

## Abstract

**Background:** Tuberculosis (TB) remains a leading cause of death among people living with HIV (PLWH), yet diagnostic methods vary in accuracy, accessibility, and implementation. Understanding how diagnostic modality influences TB detection is essential to optimizing co-infection management. **Methods:** We conducted a retrospective analysis of institutional data from Bamrasnaradura Infectious Diseases Institute (BIDI), Thailand, covering 2016–2023. TB detection rates were assessed across five diagnostic methods—chest radiography (CXR), smear microscopy, acid-fast bacilli (AFB) staining, culture, and GeneXpert MTB/RIF—relative to annual HIV-related visit volumes. **Results:** Among 56,599 HIV-related visits, TB detection rates varied substantially by diagnostic method. CXR was the most commonly used tool, detecting TB in up to 99 cases out of 6964 visits (1.42%) in 2016, though declining to 23 cases out of 6947 visits (0.33%) in 2023. GeneXpert was employed more consistently, yielding between 7 cases out of 7577 visits (0.09%) and 13 cases out of 6593 visits (0.20%) annually. Smear microscopy and AFB staining declined markedly, falling below 0.22% after 2020. These patterns reflect a gradual transition toward molecular diagnostics, which offer improved accuracy but remain underutilized in lower-tier settings. To address these gaps, we incorporated trend analyses confirming significant temporal shifts and propose a tiered TB screening framework tailored to resource availability across healthcare levels. **Conclusions:** TB detection among PLWH is strongly influenced by the diagnostic method used. Unlike HIV diagnosis—which is definitive and standardized—TB diagnosis remains fragmented and resource-dependent. Context-sensitive screening protocols are urgently needed to improve TB case detection and management, particularly in lower-level HIV care facilities.

## 1. Introduction

Tuberculosis (TB) remains one of the leading causes of death among people living with HIV (PLWH), particularly in low- and middle-income countries where the burden of co-infection is high and diagnostic resources are often limited [1]. Despite substantial global efforts to integrate TB and HIV services, timely and accurate diagnosis of TB in PLWH continues to be challenging. Contributing factors include atypical clinical presentations, immune suppression, and the inherent limitations of available diagnostic methods [2,3].

Unlike HIV, which is diagnosed through highly sensitive and standardized serological or nucleic acid-based tests, TB lacks a single definitive diagnostic tool. Accurate TB diagnosis often requires a combination of clinical judgment, radiological evaluation, and microbiological confirmation. The commonly used tools—chest radiography (CXR), sputum smear microscopy, acid-fast bacilli (AFB) staining, mycobacterial culture, and molecular techniques such as GeneXpert MTB/RIF—each offer different balances of sensitivity, specificity, accessibility, and cost-effectiveness [2,3]. These limitations are especially pronounced in immunocompromised populations like PLWH, where atypical radiological findings and paucibacillary disease are common.

Multiple studies have shown that diagnostic modality significantly influences TB prevalence estimates among HIV-positive cohorts. A study from Indonesia found that GeneXpert identified over three times more TB cases than smear microscopy [4], while a prospective cohort in South Africa revealed that up to 90% of TB cases in PLWH were subclinical and would have been missed by symptom-based screening alone [5]. In Ethiopia, Toru et al. reported varying co-infection rates depending on diagnostic method [6]. Similarly, a meta-analysis by Alanazi et al. reported a pooled HIV/TB co-infection prevalence of 16.3% (95% CI: 9.6–24.4%) across 18 studies, with a range from 0.29% in Pakistan to 44.2% in Nigeria, underscoring the heterogeneity in TB detection attributable to methodological differences [7].

Data from Asia and the Middle East add further nuance. In China, Cui et al. reported a 7.2% HIV/TB co-infection rate, with regional differences driven by diagnostic capacity and clinical protocols [8], while Gao et al. emphasized the nonspecific nature of CXR findings in PLWH and the risk of overdiagnosis without microbiological confirmation [9]. In the MENA region, Kazemian et al. highlighted underdiagnosis in settings where molecular tools remain underutilized [10]. Thai studies also echo these findings: Apidechkul et al. observed an 8.5% TB/HIV co-infection rate among northern hill-tribe populations, where access to diagnostics is limited [11], and Gatechompol et al. reported diagnostic delays in PLWH receiving care at a tertiary hospital in Bangkok [12].

Beyond high-burden countries, disparities in TB diagnosis have also been documented in low-incidence settings. For example, studies from Europe have shown longer diagnostic delays among migrants compared to native populations, illustrating that inequities in diagnostic access are a global issue rather than one confined to high-burden regions [13]. These findings emphasize that variability in diagnostic capacity contributes to inequities in TB outcomes worldwide.

Although Thailand has established a strong infrastructure for HIV treatment, empirical data on real-world TB diagnostic practices and the impact of method selection on detection outcomes remain scarce. A study conducted in northern Thailand reported a TB/HIV co-infection prevalence of 8.5% among PLWH, focusing primarily on epidemiologic factors rather than diagnostic strategies [11]. The latest WHO 2024 Global TB Report reaffirms the critical need for enhanced diagnostic capacity, especially in populations with high HIV prevalence, calling for expanded use of rapid molecular diagnostics such as GeneXpert and LAMP-based platforms at the point of care to bridge diagnostic gaps [14]. However, disparities in implementation across facility levels remain a key barrier, underscoring the urgency of context-adapted diagnostic frameworks.

In this study, we analyze eight years of HIV-related visit data from Bamrasnaradura Infectious Diseases Institute (BIDI), a Thai national infectious disease referral center. Our objectives were to (1) compare TB detection rates across different diagnostic methods; (2) examine trends in diagnostic usage over time; and (3) propose a context-sensitive TB screening framework tailored to resource availability at different healthcare levels. We also explore the diagnostic asymmetry between HIV and TB testing—a gap that complicates case detection and clinical decision-making in HIV care.

## 2. Materials and Methods

### 2.1. Study Design and Setting

This retrospective descriptive study was conducted at Bamrasnaradura Infectious Diseases Institute (BIDI), located in Nonthaburi, Thailand. BIDI is a national referral hospital under the Department of Disease Control, Ministry of Public Health, with a specialized mandate for infectious diseases. It serves as a tertiary referral center for complex cases of tuberculosis (TB), HIV, and other co-infections, and plays a critical role in national diagnosis, treatment, clinical training, and public health surveillance.

BIDI provides integrated TB/HIV services, including routine screening, diagnostic confirmation, and comprehensive treatment. This makes it an appropriate setting for evaluating patterns of diagnostic utilization and detection outcomes over time in a real-world institutional context.

### 2.2. Data Source and Population

We reviewed institutional annual summary reports from BIDI covering the period from 2016 to 2023. These reports include the total number of HIV-related clinical visits per year and the number of TB diagnoses stratified by diagnostic method. The unit of analysis was the clinical visit, not the individual patient; thus, a single patient could contribute multiple visits across or within calendar years depending on their follow-up schedule and care needs.

At BIDI, TB screening among PLWH is routinely conducted at annual visits and additional encounters when clinically indicated. Screening generally begins with the WHO four-symptom screen (W4SS), and further testing is guided by either protocol-driven requirements (e.g., radiography for annual assessment) or clinician discretion based on presenting symptoms, medical history, and resource availability. As a result, diagnostic modality selection reflects a combination of standardized algorithms and individualized clinical decision-making, influenced by infrastructure and patient factors.

This visit-based approach aligns with standard service delivery data collection practice in Thailand and offers practical insights into the volumes and type of TB diagnostic activities performed in routine HIV care. While it may overestimate the number of unique individuals, it more accurately reflects diagnostic workload, screening practices, and real-world decision-making patterns.

### 2.3. TB Diagnostic Categories

TB diagnoses were classified according to the diagnostic method used to confirm or suspect disease. Five categories were defined as follows:Chest Radiography (CXR): Considered abnormal if imaging suggested TB-related lesions, as interpreted by radiologists or attending clinicians.Sputum Smear Microscopy: Detection of acid-fast bacilli (AFB) using Ziehl–Neelsen staining on sputum samples.AFB Staining (Non-Sputum): Application of AFB staining to non-sputum specimens such as lymph node aspirates or bronchial washings.Culture: Isolation *Mycobacterium tuberculosis* from clinical specimens using solid or liquid culture media.GeneXpert MTB/RIF: A rapid, automated nucleic acid amplification test for TB DNA and rifampicin resistance.

Line Probe Assay (LPA) was recorded in institutional reports but was not considered a primary diagnostic method for TB detection. Instead, LPA is primarily used for drug resistance testing after microbiological confirmation of TB. Therefore, LPA results were excluded from comparative analyses of diagnostic yield but are noted for completeness in describing institutional diagnostic activities.

Multiple diagnostic methods may have been used during the same clinical encounter. However, the dataset captured only the number of visits associated with each method rather than patient-level linkage. As such, TB detection by diagnostic method is presented as a visit-based count and may include overlapping cases across methods.

### 2.4. Data Analysis

The primary outcome was the annual proportion of TB diagnoses by diagnostic method, calculated as the number of TB cases identified using each method divided by the total number of HIV-related visits in the corresponding year. These proportions represent method-specific diagnostic activity rather than unique TB cases, as multiple methods may have been applied during a single visit.

Temporal trends from 2016 to 2023 were visualized using bar and line graphs to highlight shifts in diagnostic utilization and yield over time. Descriptive statistics were used to summarize HIV-related visit volumes, TB detection proportions by method, and patient demographic characteristics.

To provide statistical confirmation of observed patterns, trend analyses were conducted. A Cochran–Armitage test for trend was used evaluate annual changes in TB detection proportions across the eight-year period for each diagnostic method. In addition, years were grouped into an early period (2016–2019) and a late period (2020–2023). For each diagnostic method, proportions of TB diagnoses were compared between periods using two-sample tests of proportions, with absolute differences (percentage points), 95% confidence intervals, and *p*-values reported. A two-sided α = 0.05 was used to determine statistical significance.

### 2.5. Ethics Approval

This study involved a secondary analysis of anonymized institutional data without individual identifiers. In accordance with Thai national regulations governing public health surveillance activities, informed consent was not required. The study protocol was reviewed and approved by the Institutional Review Board of Bamrasnaradura Infectious Diseases Institute, Ministry of Public Health, Thailand (IRB No. S029h-62_ExPD).

## 3. Results

Between 2016 and 2023, BIDI recorded a total of 56,599 HIV-related clinical visits. Annual visit volumes showed modest variation, ranging from a peak of 7577 visits in 2017 to a low of 6371 in 2022. A rebound was observed in 2023 with 6947 visits (Table 1).

Over the eight-year period, the proportion of female patients declined steadily from 38.8% (2699 out of 6964 visits) in 2016 to 37.0% (2570 out of 6947 visits) in 2023, indicating a consistent male predominance (approximately 62–63% annually). The mean age of patients increased from 44 years (IQR 38–51) in 2016 to 50 years (IQR 43–57) in 2023, suggesting an aging patient population that may present with evolving comorbidity and co-infection risks compared to younger cohorts (Table 1, Figure 1).

### 3.1. Diagnostic Method Use and TB Detection Patterns

Over the eight-year study period, TB diagnosis among people living with HIV (PLWH) at BIDI employed a range of diagnostic methods, each reflecting different levels of accessibility, diagnostic yield, and clinical preference.

Chest radiography (CXR) was the most frequently used method throughout all years, although its TB detection rate declined from 99 cases out of 6964 visits (1.42%) in 2016 to 23 cases out of 6947 visits (0.33%) in 2023. Its consistent use highlights both its broad availability and clinician familiarity, particularly as a rapid, non-invasive screening tool. However, CXR findings were often non-specific, and in many cases, TB diagnoses were made based solely on radiological impressions without microbiological confirmation—especially in urgent or resource-constrained settings.

GeneXpert MTB/RIF was used at modest but steady levels across the study period. Detection proportions ranged from 7 cases out of 7577 visits (0.09%) in 2017 to 14 cases out of 7080 visits (0.20%) in 2020. Annual detection remained within this narrow range, indicating stable yield despite variation in visit volumes. During the study period, the standard GeneXpert MTB/RIF platform was employed; the Ultra version was not yet in routine use.

Sputum smear microscopy and AFB staining both declined sharply in use and yield over time. In 2016, smear microscopy detected 67 cases out of 6964 visits (0.96%), whereas by 2023 only 13 cases out of 6947 visits (0.19%) were identified. Similar reductions were seen for AFB staining. These declines mirror global shifts away from smear-based diagnostics, particularly in immunocompromised populations such as PLWH, in whom smear sensitivity is reduced and extrapulmonary TB is more common.

Mycobacterial culture played a more selective role, used mainly for complex or suspected drug-resistant TB cases. Detection proportions ranged between 45 cases out of 6964 visits (0.65%) in 2016 and 17 cases out of 6947 visits (0.24%) in 2023. While used less frequently, culture remained integral to comprehensive TB care, particularly for drug susceptibility testing.

Line Probe Assay (LPA) was used infrequently during the study period, with detection proportions ranging from 4 cases out of 7577 visits (0.06%) to 12 cases out of 7498 visits (0.17%) in 2019. By 2023, LPA accounted for 5 cases out of 6947 visits (0.07%).

Together, these patterns reflect a progressive diagnostic transition at BIDI—from older, lower-sensitivity techniques such as smear microscopy toward more accurate, albeit resource-intensive, molecular diagnostics. This shift underscores both institutional capacity and the evolving clinical approach to TB/HIV co-management.

### 3.2. Temporal Shifts and Diagnostic Transitions

Over the eight-year period, a shift from traditional to modern TB diagnostics was observed at BIDI. As illustrated in Figure 1, CXR remained the most frequently used method, but its detection proportion declined after 2016.

In the period comparison (Table 2), pooling years into Early (2016–2019) and Late (2020–2023), significant declines were observed in TB detection using CXR (0.74% to 0.39%; difference −0.35; 95% CI −0.47 to −0.23; *p* < 0.001), smear microscopy (0.56% to 0.19%; difference −0.38; 95% CI −0.47 to −0.28; *p* < 0.001), AFB staining (0.56% to 0.20%; difference −0.37; 95% CI −0.46 to −0.26; *p* < 0.001), and culture (0.38% to 0.23%; difference −0.15; 95% CI −0.24 to −0.06; *p* = 0.001). By contrast, molecular methods showed no significant change between period: molecular total (0.26% vs. 0.23%; difference −0.04; *p* = 0.369), GeneXpert (0.17% vs. 0.15%; difference −0.02; *p* = 0.476), and LPA (0.12% vs. 0.10%; difference −0.02; *p* = 0.456). Overall, these findings indicate declining reliance on CXR, smear microscopy, and AFB staining, while molecular diagnostics, though used at lower levels, maintained stable yields across the study period.

### 3.3. Diagnostic Gaps and Framework Development

Across the eight-year period, the use of molecular diagnostics, particularly GeneXpert, remained modest compared to their potential diagnostic contribution, while CXR) and clinical assessment continued to account for a substantial share of TB diagnoses (Table 1, Figure 1). Despite declines, smear microscopy and AFB staining were still applied in some visits, reflecting the persistence of traditional methods in clinical workflows.

To describe potential approaches for routine TB screening in HIV care, we developed a context-sensitive framework (Table 3). The framework outlines tiered diagnostic strategies based on resource availability at three levels of care: primary/rural clinics, district hospitals, and referral centers. It incorporates CXR, GeneXpert, and other methods in varying combinations, structured to match the diagnostic capacity of each facility type.

## 4. Discussion

Our eight-year retrospective analysis of TB diagnostic methods among PLWH at a Thai national referral center reveals substantial variation in TB detection rates depending on the diagnostic modality used. The findings emphasize three overarching themes. First, diagnostic yields were highly heterogeneous across methods. CXR, smear microscopy, AFB staining, culture, and molecular assays each produced different detection proportions, reflecting the inherent strengths and weaknesses of individual tools when applied in real-world HIV care.

Second, the study underscores the fundamental methodological contrast between TB and HIV diagnostics. Whereas HIV testing relies on standardized, highly sensitive assays that provide definitive results, TB diagnosis remains fragmented and probabilistic, depending on a combination of clinical, radiological, and microbiological inputs.

Third, the results highlight the ongoing need for context-sensitive and tiered screening protocols within HIV clinical services. While global guidelines recommend molecular methods such as GeneXpert as the preferred initial test, disparities in implementation and resource availability—both within Thailand and internationally—continue to shape real-world diagnostic practices. As seen in other settings, including Europe, where migrants experience longer diagnostic delays than native populations, inequities in access to TB diagnostics are a global phenomenon that influence both case detection and outcomes.

### 4.1. TB Detection Is Strongly Method-Dependent

Our findings resonate with global evidence on the burden and variability of TB/HIV co-infection. A recent meta-analysis reported a pooled HIV/TB co-infection prevalence of 16.3% (95% CI: 9.6–24.4%) across 18 studies, with prevalence ranging from 0.29% in Pakistan to 44.2% in Nigeria [7]. This striking range mirrors the diagnostic and systemic heterogeneity we observed at BIDI. Similar patterns have been documented in Uganda [15], Eswatini [16], and Tanzania [17], where high TB incidence in PLWH underscores the influence of both local epidemiology and diagnostic infrastructure.

CXR yielded the highest TB detection proportions in our dataset, consistent with its status as a cornerstone screening tool in high-burden settings. However, its limited specificity makes it vulnerable to both over- or underdiagnosis in the absence of microbiological confirmation [2,3]. This is consistent with reports from China, where CXR-based diagnosis in PLWH has been shown to risk false positives [9].

The introduction of GeneXpert MTB/RIF represented a turning point in TB diagnostics. In our study, detection rates were modest but stable across the eight years, reflecting its superior molecular precision over smear microscopy. Evidence from Indonesia [4], South Africa [5], and China [8,18,19] consistently demonstrates that GeneXpert outperforms smear microscopy in sensitivity, particularly in immunocompromised patients. GeneXpert also provided faster turnaround times compared to culture, and studies from China further emphasized how prevalence estimates varied widely depending on diagnostic modality and regional health system capacity. Nevertheless, in our dataset GeneXpert remained underutilized, especially in earlier years and lower-level facilities. The 2024 WHO Global TB Report reaffirms this diagnostic gap, calling for universal access to rapid molecular tools as a cornerstone of TB detection in PLWH [14].

Microscopy-based methods such as sputum smear and AFB staining showed declining detection rates in our study, reflecting their globally recognized limitations. These methods have reduced sensitivity in immunosuppressed individuals and are prone to operator error in high-volume clinics. This decline is in line with WHO recommendations to prioritize molecular diagnostics over smear-based methods [1] and with reports from Germany [20], Uzbekistan [21], and Iran [10], where diagnostic strategies have similarly shifted as access to molecular platforms expanded.

Within Thailand, diagnostic heterogeneity has also been well documented. Gatechompol et al. reported a 5.4% TB/HIV co-infection prevalence in Bangkok tertiary hospital, emphasizing delays in diagnosis and treatment initiation [12], while Apidechkul observed an 8.5% prevalence among northern hill-tribe populations, where diagnostic access was constrained [11]. These national data complement our findings, reinforcing that diagnostic method selection directly shapes observed TB burden in HIV care. Improving detection requires not only expanded use of molecular tools but also deployment strategies that reflect the realities of resource availability across different health system levels.

### 4.2. HIV Diagnosis Is Definitive; TB Diagnosis Is Probabilistic

A key theme emerging from our study is the diagnostic asymmetry between HIV and TB. HIV testing—whether serological or molecular—provide near-definitive results with high sensitivity and specificity, enabling confident diagnosis and timely initiation of antiretroviral treatment. TB diagnosis, however, remains probabilistic and context-dependent, often relying on a combination of clinical suspicion, radiographic patterns, and variable microbiological tools [1,2].

This difference in diagnostic certainty has significant programmatic implications. Estimates of TB/HIV co-infection may vary depending on the diagnostic tools employed, leading to under- or over-estimation of disease burden. For example, studies from Ethiopia and Nigeria documented prevalence ranges of 16–31% and up to 44%, respectively, strongly influenced by the diagnostic methods applied [7,22]. Similar disparities have been reported in other regions, reflecting the uneven deployment of molecular diagnostics and continued reliance on smear or radiography in resource-limited settings.

Beyond prevalence estimates, diagnostic gaps directly influence patient outcomes. Delays or misclassification can contribute to ongoing transmission, delayed treatment initiation, and increased risk of adverse outcomes, including mortality. Evidence from European cohorts demonstrates that incomplete diagnostic pathways are linked to poorer prognoses and higher rates of treatment-related complications [15,20]. For PLWH, whose clinical presentation is often atypical and who are at higher risk of rapid disease progression, the consequences of diagnostic uncertainty are particularly severe.

### 4.3. Need for Context-Adapted Screening Algorithms

Current WHO guidelines recommend the use of the four-symptom screen (W4SS) at every clinical encounter, followed by molecular testing (e.g., GeneXpert) or chest radiography in symptomatic individuals [1,23]. While these recommendations are grounded in evidence, they implicitly assume that diagnostic resources are consistently available across all levels of the healthcare system—a condition that often does not hold true, particularly in rural or under-resourced settings.

Our findings indicate that real-world diagnostic practices frequently diverge from guidelines, especially where GeneXpert is limited or rationed by cost, infrastructure, or staffing. The 2024 WHO Global TB Report underscores this gap, calling for equitable expansion of rapid molecular diagnostics—including GeneXpert MTB/RIF Ultra and emerging platforms such as loop-mediated isothermal amplification (LAMP)—and their decentralization to lower-level facilities [14]. Urine LAM assays and stool-based Xpert testing are also increasingly recognized as valuable tools for improving TB detection in PLWH, particularly for extrapulmonary or paucibacillary disease.

Thai evidence further reflects these challenges. Apidechkul et al. documented an 8.5% TB/HIV co-infection rate in a northern hill-tribe populations, with higher risk among those with advanced disease and limited diagnostic access [11]. Gatechompol et al. reported diagnostic delays among PLWH in Bangkok tertiary hospitals, further highlighting gaps even in resource-rich urban centers [12]. However, these studies did not differentiate co-infection prevalence by diagnostic modality. Our analysis adds to this evidence by quantifying diagnostic yield over time, demonstrating variability across methods and underscoring the need for operational guidance adapted to resource constraints.

To bridge these disparities, we developed a descriptive, tiered TB screening framework (Table 3). This framework stratifies strategies by healthcare setting: primary/rural clinics using W4SS with structured referral and tuberculosis preventive therapy (TPT); district hospitals combining W4SS with CXR and targeted GeneXpert testing; and referral centers implementing comprehensive platforms, including GeneXpert, culture, and line probe assays. While not directly validated by patient-level outcomes, the framework integrates WHO recommendations with observed diagnostic patterns in Thailand, offering a pragmatic model for optimizing TB detection and case management in HIV clinics under variable resource conditions.

### 4.4. Strengths and Limitations

A major strength of this study lies in its eight-year scope, offering a longitudinal view of diagnostic evolution within a national reference facility. By using institutional aggregate data, the analysis captures real-world operational practices and yields policy-relevant insights into diagnostic trends over time. The setting—a high-volume, tertiary-level infectious disease hospital—adds credibility and generalizability to findings, especially for other middle-income countries with similar healthcare infrastructures.

Several limitations should be acknowledged. First, the dataset was aggregated at the visit level, rather than the patient level. This precluded the ability to link multiple visits from the same patient, censor subsequent visits after a positive TB diagnosis, or assess diagnostic overlap when more than one test was applied in the same encounter. As a result, detection proportions represent method-specific activity at the visit level and may not correspond to unique TB cases. Second, the dataset did not capture the clinical rationale for test selection (e.g., systematic screening versus symptom-driven testing), limiting interpretation of whether results reflect proactive case-finding or diagnostic evaluation of suspected disease. Third, no information was available on treatment initiation, longitudinal follow-up, or outcomes, preventing assessment of how diagnostic method influenced prognosis. Fourth, we were unable to stratify by site of disease (pulmonary versus extrapulmonary TB), an important distinction given that diagnostic yields differ substantially depending on disease presentation. This limitation may have led to underestimation of diagnostic performance for methods such as GeneXpert or culture, which are particularly relevant in extrapulmonary TB. Finally, while we proposed a tiered screening framework informed by observed patterns and international guidelines, it has not been prospectively validated with patient-level data. Future studies linking diagnostic use to clinical outcomes across care levels are needed to test and refine such models.

## 5. Conclusions

Tuberculosis detection among people living with HIV (PLWH) is strongly shaped by the diagnostic method employed. This study highlights the critical contrast between the definitive, standardized nature of HIV testing and the fragmented, resource-sensitive approach to TB diagnosis. Chest radiography remains widely used due to its accessibility but lacks specificity. GeneXpert provides greater accuracy and speed but is inconsistently deployed across levels of care. Smear-based methods continue to decline yet persist in settings with limited resources.

To close these gaps, tiered and context-adapted screening algorithms are essential for guiding TB evaluation in HIV clinics. National TB/HIV programs should move beyond one-size-fits-all protocols and adopt flexible, resource-matched strategies to ensure equitable, timely, and effective TB screening across diverse healthcare settings.

By quantifying variation in TB detection across diagnostic methods, our study provides practical evidence to support the design of responsive TB screening protocols in HIV care. These insights may help bridge diagnostic inequities and strengthen co-infection management efforts at both national and international levels.

## Figures and Tables

**Figure 1 diagnostics-15-02327-f001:**
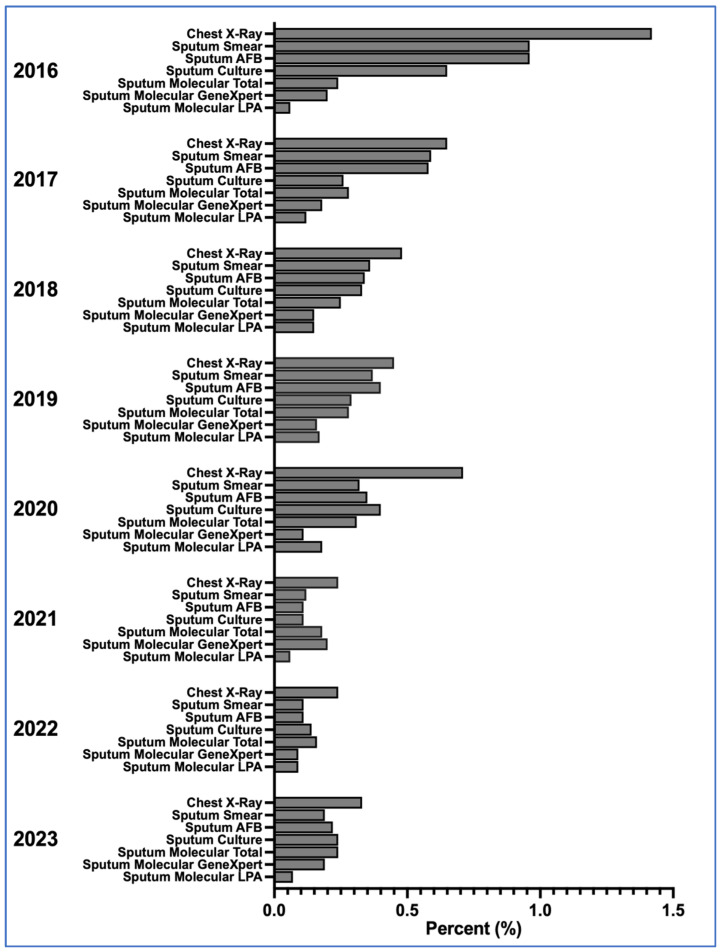
Diagnostic Method-Specific TB Detection Rates Among HIV-Related Visits at BIDI, 2016–2023. This figure illustrates the percentage of HIV-related clinical visits at Bamrasnaradura Infectious Diseases Institute (BIDI) in which tuberculosis (TB) was diagnosed using various diagnostic methods from 2016 to 2023. Diagnostic modalities include chest radiography (CXR), sputum smear microscopy, acid-fast bacilli (AFB) staining, culture, GeneXpert MTB/RIF (Cepheid, Sunnyvale, CA, USA), and line probe assay (LPA). The data show a transition away from traditional microscopy-based methods and chest X-rays toward mor consistent use of molecular diagnostics, particularly GeneXpert. The decreasing reliance on CXR for primary screening reflects evolving institutional practice and diagnostic capacity over time.

**Table 1 diagnostics-15-02327-t001:** HIV-Related Visit Characteristics and TB Diagnostic Use at BIDI (2016–2023).

Year	HIV-Related Visits	Female (%)	Mean Age (SD)	Median Age (IQR)	TB Diagnoses ^1^						
					CXR (%)	Smear (%)	AFB (%)	Culture (%)	Molecular Total (%)	GeneXpert (%)	LPA (%)
2016	6964	38.8	43.88 (10.88)	44 (38–51)	1.42	0.96	0.96	0.65	0.24	0.20	0.06
2017	7577	38.3	44.40 (10.96)	45 (38–51)	0.65	0.59	0.58	0.26	0.28	0.18	0.12
2018	7569	37.9	44.95 (11.02)	46 (38–52)	0.48	0.36	0.34	0.33	0.25	0.15	0.15
2019	7498	37.9	45.70 (11.10)	46 (39–53)	0.45	0.37	0.4	0.29	0.28	0.16	0.17
2020	7080	37.6	46.64 (11.05)	47 (40–54)	0.71	0.32	0.35	0.4	0.31	0.11	0.18
2021	6593	37.5	47.51 (11.00)	48 (41–54)	0.24	0.12	0.11	0.11	0.18	0.20	0.06
2022	6371	37.3	48.51 (10.94)	49 (42–55)	0.24	0.11	0.11	0.14	0.16	0.09	0.09
2023	6947	37.0	49.67 (10.95)	51 (43–57)	0.33	0.19	0.22	0.24	0.24	0.19	0.07

^1^ Values in TB diagnosis columns represent percentage of HIV-related visits that resulted in a TB diagnosis using that method. Multiple diagnostic methods may be used per patient.

**Table 2 diagnostics-15-02327-t002:** Period Comparison of TB Diagnostic Use at BIDI—Early (2016–2019) vs. Late (2020–2023).

Year	TB Diagnoses ^1^						
	CXR (%)	Smear (%)	AFB (%)	Culture (%)	Molecular Total (%)	GeneXpert (%)	LPA (%)
Early (%) (95% CI)	0.74 (0.64, 0.83)	0.56 (0.48, 0.65)	0.56 (0.48, 0.65)	0.38 (0.31, 0.45)	0.26 (0.21, 0.32)	0.17 (0.13, 0.22)	0.12 (0.08, 0.17)
Late (%) (95% CI)	0.39 (0.31, 0.46)	0.19 (0.14, 0.24)	0.20 (0.15, 0.25)	0.23 (0.17, 0.28)	0.23 (0.17, 0.28)	0.15 (0.10, 0.19)	0.10 (0.07, 0.14)
Difference (%) (95% CI)	−0.35 (−0.47, −0.23)	−0.38 (−0.47, −0.28)	−0.37 (−0.46, −0.26)	−0.15 (−0.24, −0.06)	−0.04 (−0.12, 0.04)	−0.02 (−0.09, 0.04)	−0.02 (−0.08, 0.04)
*p*-value	<0.001	<0.001	<0.001	0.001	0.369	0.476	0.456

^1^ Values in TB diagnosis columns represent percentage of HIV-related visits that resulted in a TB diagnosis using that method. Multiple diagnostic methods may be used per patient.

**Table 3 diagnostics-15-02327-t003:** Contextual TB Screening Framework for HIV Clinics.

Clinic Setting ^1^	Resources Available	Recommended Screening Algorithm
Primary/Rural	No radiography, no on-site GeneXpert	1. W4SS symptom screen 2. Refer symptomatic patients 3. TPT for asymptomatic after rule-out
District Hospital	CXR, limited GeneXpert	1. W4SS + CXR 2. If either positive, GeneXpert 3. Use clinical judgment if GeneXpert unavailable
Referral Center	CXR, GeneXpert, Culture, LPA	1. Routine W4SS + CXR + GeneXpert 2. Culture/LPA for resistance confirmation

^1^ This framework is intended to guide routine TB screening among people living with HIV (PLWH), tailored to the diagnostic capabilities of each healthcare setting. It integrates WHO-recommended elements and applies them contextually to optimize detection while addressing local constraints. W4SS = WHO Four-Symptom Screen (current cough, fever, weight loss, or night sweats). TPT = Tuberculosis Preventive Therapy.

## Data Availability

The datasets used and/or analyzed during the current study are available from the corresponding author upon reasonable request.

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
