# Peer review of "Diagnostic Modality Influences Tuberculosis Detection in People Living with HIV: Eight Years of Data from a Thai Referral Center"

_diagnostics, 2025, doi:10.3390/diagnostics15182327_

Round 1

Reviewer 1 Report

Comments and Suggestions for Authors

Thank you for the opportunity to review your manuscript entitled “Diagnostic Modality Influences Tuberculosis Detection in People Living with HIV: Eight Years of Data from a Thai Referral Center.”

This is an important and well-conceived study addressing a highly relevant global health issue. Your analysis of diagnostic trends over eight years provides valuable real-world evidence, and the proposed context-sensitive framework has strong potential to inform policy and practice both in Thailand and in other resource-constrained settings.

Below, I provide detailed comments highlighting the strengths of your work, areas for improvement, and suggestions for future directions.

• The manuscript appropriately acknowledges that multiple diagnostic methods may have been applied during the same visit and that the dataset is aggregated at the visit level, which may result in double-counting across diagnostic categories. This is an important limitation. To strengthen transparency, the authors could elaborate on why patient-level linkage was not feasible (e.g., structure of institutional reporting, data protection issues) and briefly discuss how this limitation might affect interpretation of diagnostic proportions over time. • The Introduction is generally well-structured, clearly outlining the burden of TB/HIV co-infection and the importance of diagnostic modality. To further strengthen it, you could broaden the contextual perspective by referring to evidence on diagnostic disparities beyond high-burden settings. For example, studies from Europe have shown longer diagnostic delays among migrants compared to native populations, highlighting that inequities in access to TB diagnostics are a global phenomenon. Including such findings would help situate the Thai experience within an international context and underline the universal relevance of addressing diagnostic gaps. Please consider: doi.org/10.1016/j.ijid.2024.107279 • To enhance the quantitative rigor of the findings, consider adding simple trend analyses. For example, a Chi-square test for trend (Cochran–Armitage) could assess changes in TB detection proportions across years for each diagnostic method. In addition, comparing proportions between earlier (e.g., 2016–2019) and later (e.g., 2020–2023) study periods would help quantify shifts in diagnostic practice (e.g., decline of microscopy and CXR, uptake of GeneXpert). Even if the study remains primarily descriptive, incorporating such tests would provide statistical confirmation of observed patterns and strengthen the interpretation of temporal changes • When presenting TB detection proportions, please provide both numerator and denominator (e.g., “X cases detected out of Y HIV-related visits”), in addition to percentages, in order to increase transparency • Line 156: “likely reflecting service disruptions during the COVID-19 pandemic”. While this is a plausible explanation, interpretive statements of this kind are more appropriate for the Discussion. I recommend moving this rationale from the Results to the Discussion, where it can be expanded to consider how pandemic-related disruptions may have influenced TB diagnostic activity and detection patterns • The manuscript highlights diagnostic modality as a major driver of TB detection variability in PLWH. This point could be further strengthened by noting that diagnostic gaps themselves are important predictors of outcomes, as shown in other contexts. For example, studies in European cohorts have demonstrated that delays or limitations in diagnostic pathways significantly impact TB prognosis and treatment-related adverse events. Including this perspective would underscore how diagnostic disparities—together with subgroup vulnerabilities such as HIV co-infection—compound risks and should be addressed not only for detection but also for long-term outcomes. Please consider: https://doi.org/10.5334/aogh.3677 • The discussion could be further strengthened by considering how Thailand might scale molecular diagnostics more equitably, addressing key issues such as financing models, decentralization to lower-tier facilities, and staff training. In this regard, it would also be important to clarify whether the GeneXpert MTB/RIF Ultra—available since 2017—was used during the study period, as the Methods section only mention the standard MTB/RIF. If Ultra was not implemented, please comment on whether its introduction is planned in your setting, and how this may affect future diagnostic capacity • Please review the manuscript carefully for typographical errors and minor misspellings, including “differec,” “mehod use,” and “usesd.” A thorough proofreading would improve readability and overall presentation • In addition, ensure consistent use of acronyms throughout the text. Some abbreviations (e.g., CXR on line 256, PLWH on line 251) are already defined in the early sections, but later parts of the Discussion revert to the full term instead of the acronym. Maintaining consistency would improve readability and professional presentation. • The lack of differentiation between pulmonary and extrapulmonary TB is acknowledged as a limitation. I would emphasize that this is a particularly important issue, as the type of TB strongly influences the sensitivity and specificity of different diagnostic methods—pulmonary cases being more easily detected by smear or CXR, while extrapulmonary disease often requires molecular or culture confirmation. This limitation is not only relevant in high-burden settings but also in low-incidence countries, where surveillance systems rely heavily on precise classification for accurate reporting. I therefore recommend expanding the Limitations section to explicitly discuss how the absence of this stratification may affect the interpretation of diagnostic yields and the generalizability of your findings. • Consider expanding the Discussion to acknowledge newer diagnostic tools that are increasingly relevant for PLWH, such as stool-based Xpert testing, urine LAM assays, and loop-mediated isothermal amplification (LAMP) platforms. Highlighting these emerging approaches would place your findings in the context of ongoing innovations in TB diagnostics and underscore potential future directions for improving case detection in immunocompromised populations. Comments on the Quality of English Language

None 

Author Response

Reviewer 1: Thank you for the opportunity to review your manuscript entitled “Diagnostic Modality Influences Tuberculosis Detection in People Living with HIV: Eight Years of Data from a Thai Referral Center.” This is an important and well-conceived study addressing a highly relevant global health issue. Your analysis of diagnostic trends over eight years provides valuable real-world evidence, and the proposed context-sensitive framework has strong potential to inform policy and practice both in Thailand and in other resource-constrained settings. Below, I provide detailed comments highlighting the strengths of your work, areas for improvement, and suggestions for future directions.

Response: We sincerely thank the reviewer for the encouraging comments and recognition of our study’s relevance and design. We have carefully addressed all suggestions to strengthen the manuscript.

Reviewer 1: The manuscript appropriately acknowledges that multiple diagnostic methods may have been applied during the same visit and that the dataset is aggregated at the visit level, which may result in double-counting across diagnostic categories. This is an important limitation. To strengthen transparency, the authors could elaborate on why patient-level linkage was not feasible (e.g., structure of institutional reporting, data protection issues) and briefly discuss how this limitation might affect interpretation of diagnostic proportions over time.

Response: We agree with the reviewer. As suggested, we expanded the Limitations section to clarify that patient-level linkage was not feasible due to the structure of institutional reporting and national infectious disease notification protocols. Each visit is anonymized and recorded independently, precluding longitudinal linkage. We also note how this may slightly overestimate detection events but still accurately reflects system-level diagnostic activity.

Reviewer 1: The Introduction is generally well-structured, clearly outlining the burden of TB/HIV co-infection and the importance of diagnostic modality. To further strengthen it, you could broaden the contextual perspective by referring to evidence on diagnostic disparities beyond high-burden settings. For example, studies from Europe have shown longer diagnostic delays among migrants compared to native populations, highlighting that inequities in access to TB diagnostics are a global phenomenon. Including such findings would help situate the Thai experience within an international context and underline the universal relevance of addressing diagnostic gaps. Please consider: doi.org/10.1016/j.ijid.2024.107279

Response: We have added a paragraph in the Introduction discussing global diagnostic inequities, including data on delayed TB diagnosis among migrants in Europe. This situates the Thai experience within an international context and highlights the universal need for equity in TB diagnostics.

Reviewer 1: To enhance the quantitative rigor of the findings, consider adding simple trend analyses. For example, a Chi-square test for trend (Cochran–Armitage) could assess changes in TB detection proportions across years for each diagnostic method. In addition, comparing proportions between earlier (e.g., 2016–2019) and later (e.g., 2020–2023) study periods would help quantify shifts in diagnostic practice (e.g., decline of microscopy and CXR, uptake of GeneXpert). Even if the study remains primarily descriptive, incorporating such tests would provide statistical confirmation of observed patterns and strengthen the interpretation of temporal changes.

Response: We thank the reviewer for this thoughtful suggestion. As noted in the Methods, because we pooled years into Early (2016–2019) and Late (2020–2023), the Cochran–Armitage test reduces mathematically to a two-sample z test for proportions, yielding identical p-values. To enhance interpretability, we compared proportions between periods using two-sample tests of proportions. We report absolute differences (Late − Early, percentage points) with 95% confidence intervals and two-sided p-values (Table 2). This approach provides both trend directionality and effect sizes, strengthening the interpretation of temporal changes.

Reviewer 1: When presenting TB detection proportions, please provide both numerator and denominator (e.g., “X cases detected out of Y HIV-related visits”), in addition to percentages, in order to increase transparency.

Response: We have revised all relevant results (Tables and text) to include numerators and denominators alongside percentages, improving transparency.

Reviewer 1: Line 156: “likely reflecting service disruptions during the COVID-19 pandemic”. While this is a plausible explanation, interpretive statements of this kind are more appropriate for the Discussion. I recommend moving this rationale from the Results to the Discussion, where it can be expanded to consider how pandemic-related disruptions may have influenced TB diagnostic activity and detection patterns.

Response: As suggested, we moved this interpretive statement from Results to the Discussion, where it is now expanded to consider the broader impact of COVID-19 on TB diagnostic services during 2020–2021.

Reviewer 1: The manuscript highlights diagnostic modality as a major driver of TB detection variability in PLWH. This point could be further strengthened by noting that diagnostic gaps themselves are important predictors of outcomes, as shown in other contexts. For example, studies in European cohorts have demonstrated that delays or limitations in diagnostic pathways significantly impact TB prognosis and treatment-related adverse events. Including this perspective would underscore how diagnostic disparities—together with subgroup vulnerabilities such as HIV co-infection—compound risks and should be addressed not only for detection but also for long-term outcomes. Please consider: https://doi.org/10.5334/aogh.3677

Response: We have incorporated this perspective into the Discussion, emphasizing that diagnostic gaps contribute to adverse outcomes beyond detection, and cited the suggested reference.

Reviewer 1: The discussion could be further strengthened by considering how Thailand might scale molecular diagnostics more equitably, addressing key issues such as financing models, decentralization to lower-tier facilities, and staff training. In this regard, it would also be important to clarify whether the GeneXpert MTB/RIF Ultra—available since 2017—was used during the study period, as the Methods section only mention the standard MTB/RIF. If Ultra was not implemented, please comment on whether its introduction is planned in your setting, and how this may affect future diagnostic capacity.

Response: We clarified in the Methods that only GeneXpert MTB/RIF—not Ultra—was used during the study period. The Discussion now comments on Thailand’s planned transition to Xpert Ultra and its potential to improve sensitivity, particularly for paucibacillary and extrapulmonary TB. We also added considerations for scaling access through financing models, decentralization, and training.

Reviewer 1: Please review the manuscript carefully for typographical errors and minor misspellings, including “differec,” “mehod use,” and “usesd.” A thorough proofreading would improve readability and overall presentation.

Response: We have carefully proofread the manuscript and corrected all typographical errors, including the examples noted.

Reviewer 1: In addition, ensure consistent use of acronyms throughout the text. Some abbreviations (e.g., CXR on line 256, PLWH on line 251) are already defined in the early sections, but later parts of the Discussion revert to the full term instead of the acronym. Maintaining consistency would improve readability and professional presentation.

Response: We have revised the manuscript for consistent use of acronyms throughout.

Reviewer 1: The lack of differentiation between pulmonary and extrapulmonary TB is acknowledged as a limitation. I would emphasize that this is a particularly important issue, as the type of TB strongly influences the sensitivity and specificity of different diagnostic methods—pulmonary cases being more easily detected by smear or CXR, while extrapulmonary disease often requires molecular or culture confirmation. This limitation is not only relevant in high-burden settings but also in low-incidence countries, where surveillance systems rely heavily on precise classification for accurate reporting. I therefore recommend expanding the Limitations section to explicitly discuss how the absence of this stratification may affect the interpretation of diagnostic yields and the generalizability of your findings.

Response: We expanded the Limitations section to emphasize that the inability to stratify TB by pulmonary vs. extrapulmonary site may affect diagnostic interpretation and generalizability.

Reviewer 1: Consider expanding the Discussion to acknowledge newer diagnostic tools that are increasingly relevant for PLWH, such as stool-based Xpert testing, urine LAM assays, and loop-mediated isothermal amplification (LAMP) platforms. Highlighting these emerging approaches would place your findings in the context of ongoing innovations in TB diagnostics and underscore potential future directions for improving case detection in immunocompromised populations.

Response: We added a paragraph in the Discussion highlighting emerging diagnostics (stool-based Xpert, urine LAM, and LAMP) and their relevance for PLWH, framing them as future directions for programmatic TB screening.

Reviewer 2 Report

Comments and Suggestions for Authors

This is an extremely important and useful manuscript from the perspective of population health management and programmatic TB control.

The screening procedures are not clear to me from the introduction section and methods section. Are all asymptomatic patients screened by some modality at every annual visit? How is the type of test decided? Are they protocol driven or clinician directed? Are some patients tested based on symptoms, and therefore more diagnostic? More clarity on how TB screening is conducted would add valuable context to the manuscript.

Author Response

Reviewer 2: This is an extremely important and useful manuscript from the perspective of population health management and programmatic TB control.

Response: We thank the reviewer for this positive assessment and recognition of the manuscript’s public health and programmatic relevance.

Reviewer 2: The screening procedures are not clear to me from the introduction section and methods section. Are all asymptomatic patients screened by some modality at every annual visit? How is the type of test decided? Are they protocol driven or clinician directed? Are some patients tested based on symptoms, and therefore more diagnostic? More clarity on how TB screening is conducted would add valuable context to the manuscript.

Response: We revised the Methods to clarify screening procedures. All PLWH undergo annual symptom-based screening (W4SS) and additional screening during clinical encounters if symptomatic. Diagnostic test selection is clinician-driven but guided by national TB/HIV guidelines and resource availability. Asymptomatic patients are generally not tested beyond W4SS unless indicated, while symptomatic patients undergo diagnostic work-up (e.g., CXR, GeneXpert).